# A Na,K-ATPase–Fodrin–Actin Membrane Cytoskeleton Complex is Required for Endothelial Fenestra Biogenesis

**DOI:** 10.3390/cells9061387

**Published:** 2020-06-03

**Authors:** Meihua Ju, Sofia Ioannidou, Peter Munro, Olli Rämö, Helena Vihinen, Eija Jokitalo, David T. Shima

**Affiliations:** 1Translational Vision Research, UCL Institute of Ophthalmology, London EC1v 9EL, UK; meihua_ju@yahoo.com; 2Edmond de Rothschild Group, 75401 Paris, France; s.ioannidou@gmail.com; 3Electron Microscopy Unit, UCL Institute of Ophthalmology, London EC1v 9EL, UK; peter.munro@ucl.ac.uk; 4Cell and Molecular Biology Program, University of Helsinki, 00014 Helsinki, Finland; ojramo@gmail.com (O.R.); eija.jokitalo@helsinki.fi (E.J.); 5Electron Microscopy Unit, Institute of Biotechnology, University of Helsinki, 00014 Helsinki, Finland; helena.vihinen@helsinki.fi

**Keywords:** fenestrae, moesin, annexin II, actin, fodrin, Na,K-ATPase, submembrane cytoskeleton

## Abstract

Fenestrae are transcellular plasma membrane pores that mediate blood–tissue exchange in specialised vascular endothelia. The composition and biogenesis of the fenestra remain enigmatic. We isolated and characterised the protein composition of large patches of fenestrated plasma membrane, termed sieve plates. Loss-of-function experiments demonstrated that two components of the sieve plate, moesin and annexin II, were positive and negative regulators of fenestra formation, respectively. Biochemical analyses showed that moesin is involved in the formation of an actin–fodrin submembrane cytoskeleton that was essential for fenestra formation. The link between the fodrin cytoskeleton and the plasma membrane involved the fenestral pore protein PV-1 and Na,K-ATPase, which is a key regulator of signalling during fenestra formation both in vitro and in vivo. These findings provide a conceptual framework for fenestra biogenesis, linking the dynamic changes in plasma membrane remodelling to the formation of a submembrane cytoskeletal signalling complex.

## 1. Introduction

The precise regulation of blood–tissue interchange is critical for integration of the microvasculature and virtually all organ systems. Accordingly, endothelial cells adopt highly specialised features to mediate the exchange of fluids and macromolecules across the vascular wall [1]. For example, the blood–brain barrier consists of fortified cellular junctions which strictly restrict access of macromolecules to neural tissue [2]. At the other extreme are endothelial fenestrae, transcellular plasma membrane pores that passively mediate bidirectional blood–tissue exchange in the microvasculature of numerous organs, primarily with endocrine, absorptive, or filtrating functions [3]. Fenestrae are assumed to be central to many essential functions of the vasculature, such as the production of primary urine from blood [4], the sieving of lipoprotein particles [5], and the rapid vascular access and systemic dissemination of hormones from endocrine organs [6], though little direct experimentation exists. Ectopic formation of endothelial fenestrae is also implicated in tissue oedema characteristic of several pathological states, such as in certain tumours or in diabetic retinopathy [7,8].

Ultrastructurally, a single fenestral pore is ~60–70 nm in diameter and spans the entire thickness of the cell without disrupting the continuity of the plasma membrane [9]. The high degree of plasma membrane curvature at the rim of the fenestral pore [10] suggests that molecular components must exist to exert force on the membrane to ensure its deformation. In most vascular beds, the pore contains a pinwheel-like diaphragm that is formed by the protein PV-1, which is the only known constituent of fenestrae but is also present in the diaphragm of endothelial caveolae and plasma membrane channels [11]. PV-1 is not required for fenestra biogenesis but rather is involved in regulating fenestra organisation and pore size [12]. Fenestrae number in the hundreds per cell and are organised in clustered arrays demarcated by microtubules and termed sieve plates (Figure 1A), which occur in regions of the cell periphery where apical–basal plasma membranes are positioned less than 40 nm apart (Figure 1B). The extreme attenuation of the cell means that most organelles and cytoskeletal elements are excluded from fenestral sieve plates, yet despite the apparent lack of underpinning cyto-architecture, fenestral pores are highly organised in linear arrays (Figure 1C). No structural or mechanistic framework exists for explaining this organisation.

In contrast to the abundant knowledge of endothelial caveolae biology, fenestral pore and sieve plate cyto-architecture and biogenesis remain elusive. The primary reasons for the lack of information more than 60 years after their initial description include a lack of cellular models, the lack of appropriate markers for fenestrae, and the heavy reliance on laborious ultrastructural methods. We have previously described an endothelial cell model whereby fenestrae are induced by actin microfilament disassembly [12]. In the cell system, VEGF-A, which has been shown to induce fenestrae formation in vivo, triggers a modest increase in fenestra numbers that is dependent on localised disassembly of the actin cytoskeleton, whereas a 10-fold higher density of fenestrae were rapidly induced using the actin depolymerising agent latrunculin A (LtA). This latter condition produces quantities of fenestrae sufficient for biochemical studies, and, using this model of fenestra biogenesis, we have performed subtractive proteomics. Counterintuitively, we found several actin-binding proteins enriched in fenestrated plasma membranes, including the FERM protein moesin and annexin II. Using a combination of correlative light–electron microscopy, electron microscopy, biochemistry, and functional studies we show that these actin-binding proteins differentially regulate the assembly of an actin–fodrin submembrane cytoskeleton that is present in the sieve plate and essential for fenestra biogenesis. This cytoskeleton directly links to the fenestral pore through interaction with PV-1 and the Na,K-ATPase, which acts as a negative regulator of fenestra formation.

## 2. Materials and Methods

### 2.1. Chemicals and Antibodies

All standard chemicals were obtained from Sigma (Dorset, UK) unless otherwise indicated. The antibodies used for this study are the following: PV-1 antibody was from Covalab (Cambridge, UK). Antibodies to moesin, phosphorylated moesin, and ezrin were from Cell Signalling (London, UK); antibody to annexin II was from Santa Cruz (Dallas, TX, USA); antibodies to β-tubulin and β-actin were from Sigma; antibodies to CD31 and fodrin were from Abcam (Cambridge, UK); and antibody to Na,K-ATPase was from Millipore (Feltham, UK).. AlexaFluor secondary antibodies were from Invitrogen (Paisley, UK), 1.4 nm nano-gold conjugated secondary antibodies from were Nanoprobes (Buckingham, UK), and Amersham ECL HRP-conjugated secondary antibodies were from Invitrogen (Paisley, UK). F-actin was visualised with AlexaFluor phalloidin (Invitrogen, Paisley, UK).

Antibodies against cofilin, filamin-1, paralemmin, transgelin, myosin IIa and IIb, reticulon 2, atlastin, arp3, talin, vinculin, enolase, putative RNA-binding protein 3, and NM-23 (nucleoside diphosphate kinase) were from Santa Cruz; antibody against merlin was from Cell Signalling; antibodies against reticulons 3 and 4, EEA-1, SAR1, calreticulin, calnexin, rab5, DP-1, KDEL, lamp1, VE-cadherin, desmin, CD44, aquaporins 1 and 4, and hnRNP-K (heterogenous nuclear ribonucleoprotein K) were from Abcam; antibody against FAK (focal adhesion-associated kinase) was from BD Biosciences (Wokingham, UK); antibodies against caveolin, rab4, integrins, and IRS p58/p53 were from BD Transduction; antibody against vimentin was from BD Pharmingen (Wokingham, UK); antibodies against radixin, merlin, syntaxins 1–4 and 5A, ERGIC-53, sec23, and sec31A were from Sigma.

### 2.2. Cell Culture and Fenestra Induction

The endothelioma cell line bEND5 (kindly provided by Dr. Britta Englehardt, University of Bern, Switzerland) was maintained in DMEM with high glucose (4.5 g/L) (Invitrogen) containing 10% FBS and antibiotics. Twenty-four hours prior to fenestra induction, cells were seeded at a density equivalent to 1.5 × 106 cells per 100 mm dish on gelatine-coated coverslips, formvar grids, or culture dishes. Cells were induced by using 1.25 µM of latrunculin A (Sigma) for 3 h, 75 ng of recombinant rodent VEGF-164 (kindly provided by Dr. Dominik Krilleke, University College London, UK) or 200 μM of ouabain (Sigma) for 24 h.

### 2.3. Immunocytochemistry and Confocal Imaging

Cells grown on coverslips were fixed at −20 °C for 7 min or 4% paraformaldehyde (PFA) for 10 min. PFA-fixed cells were permeabilised with 0.1% Triton X-100 in PBS and blocked with 10% serum in PBS. Indicated primary antibodies and corresponding secondary antibodies were serially added. Fluorescent images were captured with laser scanning confocal microscope (LSM700 with an LSM-TPMT camera, Zen 2009 software; Carl Zeiss (Cambridge, UK)) using a 63×, 1.40 NA oil objective (Carl Zeiss) for whole-mount images. All images were acquired at room temperature. Images shown were processed by Zen 2009 lite software (Cambridge, UK) to enhance visibility by adjusting brightness, contrast, and levels, applying pseudocolouring where appropriate. Within an individual figure, all images were subjected to the same post-acquisition processing.

### 2.4. SiRNA Transfection and Correlative Light–Electron Microscopy

Cells were transfected with Lipofectamine RANi MAX (Invitrogen) in culture medium without antibiotics. Silencer Select Pre-designed siRNAs (Ambion, Loughborough, UK) against moesin (ID s70074), radixin (ID s201923), annexin II (ID s63242), fodrin (ID s232645), and Silencer Select Negative Control siRNA (#1) were used at a final concentration of 5 nM. The knockdown and cell phenotype were assessed 24, 48, and 72 h later.

Cells were grown on numbered grids, transfected with siRNA, and induced to form fenestrae using the standard protocol. Cells were fixed with 4% PFA/0.1% glutaraldehyde for 15 min and then were immunolabelled. Cells with no detectable immunostaining for the target protein were identified under an epifluorescence microscope, and their position was recorded. After processing for TEM, the identified cells were re-located under electron microscopy and imaged as described in Section 2.6.

### 2.5. Western Blot Analysis and Immunoprecipitation

Whole-cell protein lysates were made from cells solubilised in medium stringency lysis buffer (0.025 M Tris, 0.15 M NaCl, 0.001 M EDTA, 1% NP-40, 5% glycerol, pH 7.4) and freshly added protease and phosphatise inhibitor cocktail (Thermo Scientific, Loughborough, UK). The resulting lysates were subjected to Western blotting with primary antibodies followed by HRP-conjugated secondary antibodies (Thermo Scientific) and developed using Amersham ECL kit (Invitrogen). For immunoprecipitation, lysate with 500 µg of total protein was precleared with Protein A/G Agarose (Thermo Scientific) and then incubated with PV-1, moesin, fodrin, or Na,K-α antibodies overnight at 4 °C. After a further 4 h incubation with Protein A/G Agarose, beads were washed (0.025 M Tris, 0.15 M NaCl, pH 7.2), and the eluted immune complexes were analysed by immunoblotting.

### 2.6. Electron Microscopy

For wholemount TEM, cells grown on formvar grids were fixed in 2.5% PFA/1.25% glutaraldehyde (Electron Microscopy Sciences, Wisbech, UK) in 0.1 M sodium cacodylate buffer, postfixed in 1% Osmium tetroxide (Agar Scientific Ltd. Stansted, UK), dehydrated, and dried in hexamethyldisalazane. For thin-section TEM, cell monolayers and animal tissue samples were fixed in a mixture of 3% glutaraldehyde and 1% PFA in 0.08 M sodium cacodylate buffer. Specimens were processed and then embedded in araldite. Ultrathin sections were cut and stained with Reynold’s lead citrate. Wholemount cell specimens and stained ultrathin sections were examined in a JEOL 1010 TEM operating at 80 kV, and images were recorded using a Gatan Orius B digital camera and Digital Micrograph (Leicester, UK).

For pre-embedding immuno-EM, cells were fixed with PLP-fixative (2% formaldehyde in 0.01 M periodate, 0.075 M Lysine-HCl, 0.037 M sodium phosphate buffer, pH 7.4) and then were incubated with primary and nano-gold conjugated secondary antibodies. Specimens were processed for TAAB embedding resin, as previously described [13]. For electron tomographic analysis, bEND5 cells were grown on 1.5-mm sapphire discs coated with gelatine. After 1 h treatment with 1.25 µM LtA, the cells were covered with 20% BSA in DMEM (BioWhittaker, Wokingham, UK) and high-pressure frozen with Leica EM Pact device (Leica Microsystems, Cambridge, UK). Substitution was carried out in 0.3% uranyl acetate (Ted Pella, Inc., Stansted, UK), 2% OsO_4_, and 10% water in ethanol using a freeze substitution device (AFS2, Leica Microsystems). Specimens were first kept at −90 °C for 16 h, after which the temperature was increased by 10 °C/h to 0 °C. The specimens were then washed with cold ethanol and acetone, gradually infiltrated into Epon at room temperature (acetone:Epon at 1:1, 0:1, 0:1, 30 min each) and flat embedded. Sections 90 nm thick were cut parallel to the sapphire disc and imaged using Tecnai FEG 20 microscope (FEI Corp. Cambridge, UK) operated at 200 kV. Images were collected with a 4k × 4k Ultrascan 4000 CCD camera (Gatan Corp. Leicester, UK) at 19,000×, providing a 2× binned pixel size of 1.2 nm. For a tilt series, the specimen was tilted at 1° intervals using a high-tilt specimen holder (model 2020; E.A. Fischione Instruments, Surrey, UK) between ±62°. The images were acquired by Serial EM software [14], and dual-axis tomography was applied [15]. The alignment of tilt series and tomographic reconstruction were done with IMOD software package [16] using 10-nm colloidal gold particles overlaid on the top of the section as fiducial markers. Modelling was done with Amira software (TGS Inc. Surrey, UK).

Fenestrated membrane area was quantified from wholemount TEM images of 50–100 transfected cells in each treatment group using threshold setting in Image J software (National Institutes of Health, Bethesda, MD, USA).

Fenestral density was quantified from high-magnification TEM images (1000×) using Image Pro-Plus 6.1 software (National Institutes of Health, Bethesda, MD, USA).

### 2.7. Animal Studies

Male Sprague Dawley rats (250–300 g) were used throughout the in vivo study. All procedures were in accordance with Home Office standards and were reviewed by an institutional animal care committee (PIL 70/21662, 29 October 2008). Modified from the previously described methods [17], rats were anesthetised with a mixture of ketamine (75 mg/kg body weight) (Fort Dodge Animal Health, Fort Dodge, IA, USA) and xylazine (25 mg/kg body weight) (IVX Animal Health, Miami, FL, USA). The cremaster muscle was surgically exposed, and 100 µL of PBS, recombinant rodent VEGF-164 (75 ng in 100 μL of PBS), or ouabain (200 µM, 100 μL) was topically applied. To detect and localise vascular leakage, the femoral vein was exposed and used to inject 10 mg/kg body weight of lysine-fixable 3KD FITC-dextran (Invitrogen). After 10 min, the cremaster was fixed in situ for 4 min with 4% PFA, then surgically removed, quickly immersed in 4% PFA, pinned flat on a piece of cork board, and fixed overnight in 4% PFA. The specimen was mounted in mounting medium for epifluorescence imaging. After washing out the mounting medium, the specimen was trimmed under a dissection epifluorescence microscope and processed for thin-section TEM.

To calculate the frequency of fenestrated vasculature, vessels in the most superficial layer of cremaster tissue were counted. Three well-preserved sections in close proximity were counted, and the average frequency was calculated; the final result was calculated from at least three rats from independent experiments. For each fenestrated vasculature, the vessel perimeter length, the fenestrated vessel length, and the number of fenestrae were determined.

### 2.8. Statistical Analysis

All the data were analysed by two-tailed Student’s *t*-test. All experiments were performed at least three times. Values are presented as means ± SEM.

## 3. Results

### 3.1. Actin-Binding Proteins are Enriched in Fenestrated Endothelial Plasma Membranes

The composition of fenestrae and sieve plates was probed using a subtractive proteomic approach. We focused on the observation that LtA treatment could induce abundant fenestrae in the bEND5 endothelioma cell line [12]. Treatment with LtA could also re-induce fenestrae in primary mouse liver sinusoidal endothelial cells that had dedifferentiated and had previously lost all fenestrae. However, the endothelioma line provided a more abundant source for isolation of fenestrated membranes. For example, treatment of cells with LtA produced high fenestra densities (~3/µm^2^ plasma membrane) and a greater than 100-fold induction relative to untreated cells, in which fenestrae were virtually absent (Figure 1D). As described previously [12], in LtA-treated bEND5 cells, fenestral sieve plates were readily distinguishable by PV-1 immunopositive patches at the cell periphery, delineated by microtubules (Figure 1E).

A silica-based affinity isolation method was chosen in order to provide highly enriched plasma membranes while maintaining the structural integrity of membranous microdomains such as caveolae [18]. With this approach, we achieved an approximately 20-fold enrichment for plasma membrane (Figure 1F). Two-dimensional gel electrophoresis and mass spectrometry were used to compare the plasma membrane protein profiles of induced and noninduced endothelioma cells. From these profiles, we identified >40 proteins assigned to the induced state, with ~25% being classified as actin-binding proteins (Figure 1G). This was unexpected, since the basis of fenestra induction relied upon disassembly of the actin microfilament-based cytoskeleton. However, based on an initial screening of proteomic candidates by immunostaining, two proteins were prioritised for further investigation—the actin- and phosphoinositide/membrane-binding proteins moesin and annexin II.

### 3.2. Moesin and Annexin II Are Present in Fenestral Sieve Plates

ERM proteins are a family of membrane–cytoskeleton adaptors comprised of ezrin, radixin, and moesin, along with the more distantly related merlin. The core ERM proteins are highly homologous, and functional redundancy has been reported between family members in vitro [19]. Immunolocalisation in bEND5 cells confirmed colocalisation of moesin with PV-1 in fenestral sieve plates (Figure 2A). In contrast, ezrin was excluded from sieve plates and was associated with the adjacent microtubule- and organelle-rich regions (Figure 2B). Thus, despite similar roles, ezrin appears to be associated with microtubule-rich regions, whilst moesin sits in fenestra-rich regions. Annexin II is a membrane-associated actin-binding protein that has been linked to organelle biogenesis and trafficking, particular within the endosomal compartment [20]. In contrast to moesin, annexin II did not colocalise with PV-1 within sieve plates; rather, it was found both associated with the microtubule-rich regions and also in small islands within the sieve plates that in many cases appeared to be contiguous with the organelle-rich areas (Figure 2C). The immunostaining observed does suggest endosomal location, but this needs further follow-up studies. We also screened for the cellular distribution of a large number (>25; see Appendix A) of cytoskeletal and membrane-associated proteins in fenestrated bEND5 cells and found none distributed in a similar manner to moesin and annexin II.

Because of the striking colocalisation with PV-1, we first focused on moesin. Prior to induction of fenestrae with LtA, moesin was broadly distributed along the plasma membrane and not colocalised with PV-1, which is primarily present in caveolae. Upon treatment with LtA, PV-1 redistributed, along with moesin, into fenestral sieve plates (Figure 2D). Immunoprecipitation analysis suggested a newly established association between moesin and PV-1 in fenestrated cells (Figure 2E). Immunoelectron microscopy on thin sections of bEND5 cells confirmed that moesin was highly enriched in sieve plates (Appendix A). Within the sieve plate, moesin was found adjacent to fenestral pores, often in electron-dense patches, suggestive of an underlying structure (Appendix A, right panel). A cross-section view in nonfenestrated regions demonstrated that moesin was enriched under the plasma membrane (Appendix A, bottom panel), as previously reported [21]. Activation of moesin’s actin-binding function and its ability to trigger filopodia formation are associated with threonine phosphorylation [22,23]. Although moesin phosphorylation levels were unchanged in fenestrated cells (Appendix A), moesin in the fenestral sieve plate is phosphorylated, similar to moesin found in plasma membrane microspikes in control bEND5 cells (Appendix A).

### 3.3. Differential Regulation of Fenestra Biogenesis by Moesin and Annexin II

Correlative light–electron microscopy following exposure to siRNA was used to determine if moesin and annexin II play a functional role in fenestra formation. Cells were grown on numbered grids, transfected with siRNA, and induced to form fenestrae using the standard protocol. After 3 h induction, cells with no detectable immunostaining for moesin or annexin II were identified and PV-1 immunostaining was documented, followed by wholemount TEM. Following moesin knockdown (Figure 3A), PV-1 was no longer present in peripheral sieve-plate-like structures but rather was colocalised with the bulk of cellular organelles, associated with microtubules, suggesting a block in fenestra formation (Figure 3B). Analysis of grids by EM and quantification confirmed the knockdown led to >90% loss in membrane fenestration (Figure 3E). In contrast, annexin II knockdown led to the opposite cellular phenotype, an apparent increase in PV-1 in the sieve plate area, which EM confirmed corresponded to an increase in sieve plate area (Figure 3D,E). Moreover, in addition to increasing fenestrated sieve plate area, annexin II knockdown also increased the density of fenestrae in individual sieve plates (Figure 3F). In summary, these data suggest that moesin is required for fenestra formation, whilst annexin II plays a negative regulatory role.

### 3.4. An Actin/Fodrin-Based Membrane Cytoskeleton Is Required for Fenestra Biogenesis

The results presented above implicating a role for two actin-binding proteins led us to further examine the potential role for an actin cytoskeleton in fenestra formation. First, we further scrutinised F-actin distribution in fenestrated cells. We have previously shown that sieve plates form in areas devoid of actin stress fibres in cells induced to form fenestrae with VEGF-164 or LtA [12]. However, using the F-actin probe phalloidin, we observed that following LtA treatment, distinct F-actin positive regions remained and colocalised with moesin and PV-1 in sieve plates (Figure 4A). The F-actin within sieve plates was of much lower fluorescence intensity than stress fibres and did not label with DNAse I, a globular actin-binding protein, suggesting that the actin was indeed in filamentous form, albeit very short filaments. The presence of actin cytoskeleton around fenestrae has been previously suggested in LtA-treated liver endothelial cells, but its structure and function remain poorly understood [24]. We turned to electron tomography, a technique capable of providing 3D data for reconstruction of detailed subcellular structures, in an attempt to visualise cytoskeletal elements associated with fenestral sieve plates in our cell system. Tomographic projections from chemically fixed cells with fenestrae yielded no discernible cytoskeletal structures in the sieve plates (Figure 4B and Video S1A); however, projections from high-pressure frozen and freeze substituted cells consistently revealed intertwining cytoskeletal fibres that tracked both proximal and parallel to the linear arrays of fenestral pores in sieve plates (Figure 4C and upper and lower insets on right; also see Video S1B). This cytoskeletal scaffold was present in all sieve plates observed and had an organisation that could fulfil the presumptive need to organise the fenestral pores and stabilise the large, attenuated, and perforated sheets of plasma membrane. The interlaced appearance of the network, and our data on the presence of short F-actin microfilaments in the sieve plates, led us to explore the association of fenestrae with the spectrin membrane cytoskeleton.

Spectrin was first identified in erythrocytes [19] and was subsequently identified in nonerythroid cells [25,26], where it is referred to as fodrin. The spectrin membrane skeleton is localised immediately under the plasma membrane and enables membrane distensibility and fortification for cells, such as erythrocytes, in rolling and adapting their shape as they travel through the narrowest capillaries [27,28]. The nonerythroid membrane skeleton is primarily composed of fodrin α- and β-heterotetramers, ankyrin, protein 4.1, and, importantly, a short actin protofilament [29,30] which is not affected by actin depolymerising drugs presumably due to the capping by spectrin and protein 4.1 [31,32]. Immunostaining revealed that following fenestra induction, alpha fodrin was relocated into the newly formed PV-1-positive sieve plates (Figure 5A). Immunoprecipitation experiments confirmed a dynamic association of fodrin and β-actin, with a transient decrease in co-association of the two cytoskeletal components, which was re-established at 30 min, corresponding to the time that definitive fenestrated sieve plates begin to become detectable (Figure 5B). This suggests that following disassembly of actin microfilaments, there is a re-association of actin and fodrin, presumably in the form of a fodrin-bound protofilament within the membrane skeleton. The importance of this cytoskeletal complex to fenestra formation was underscored by depletion of alpha fodrin using siRNA (Figure 5C). Similar to observations in moesin knockdown cells, immunostaining (Figure 5D) and wholemount-TEM micrographs (Figure 5E, left panel) showed that fodrin-deficient cells typically lacked an attenuated sieve-plate structure located between the microtubule/organelle-rich arbours. Quantification confirmed an approximate 10-fold decrease in fenestrated plasma membrane following alpha fodrin knockdown (Figure 5E, right panel).

### 3.5. Assembly of a Fenestra/Na,K-ATPase/Membrane Cytoskeleton Complex Is Regulated by Moesin and Annexin II

Having identified a putative cytoskeletal scaffold, we sought to determine the link to the fenestral pore. The fodrin-based membrane cytoskeleton associates with transmembrane proteins which serve to anchor the plasma membrane to the underlying support lattice [33,34]. One of these is the Na,K-ATPase, or sodium pump, a highly-conserved integral membrane protein that is expressed in virtually all cells of higher organisms and belongs to the P-type ATPase superfamily [35,36]. Immunostaining of the α subunit of the Na,K-ATPase confirmed an enrichment of the Na,K-ATPase in fenestral sieve plates (Figure 6A); immunogold labelling and TEM imaging suggested that the pump was present on the rim of the pore (Figure 6B). Immunoprecipitation of the Na,K-ATPase α subunit confirmed the localisation data, showing association with PV-1 and fodrin, and levels of co-association much greater in fenestrated cells than in control cells (Figure 6C). These findings strongly suggested that PV-1 and the Na,K-ATPase are involved in linking the fenestral pore to the fodrin cytoskeleton. Importantly, the assembly of the fenestra–cytoskeleton complex was differentially regulated by moesin and annexin II. Co-immunoprecipitation of fodrin and PV-1 was abolished in LtA-induced cells transfected with moesin siRNA but was enhanced following transfection of annexin II siRNA (Figure 6D).

### 3.6. The Na,K-ATPase Is a Negative Regulator of Fenestra Formation In Vitro and In Vivo

Although it is often viewed solely as a sodium pump, Na,K-ATPase has been shown to have numerous ion-transport-independent roles in the last decade, including roles in cell junction formation, cell migration, and signal transduction [37,38]. The general consensus is that the Na,K-ATPase carries out these nonpump roles acting primarily as a plasma membrane scaffold regulating the availability of several important signalling molecules, such as Src [39], PI 3-kinase [40], adaptor protein2 [41], and PLC-γ [42]. To explore a regulatory role for Na,K-ATPase, we treated bEND5 cells with ouabain, a cardiac glycoside used clinically to bind and inhibit the catalytic function of the Na,K-α subunit [43]. Immunostaining for PV-1 revealed that ouabain was able to induce formation of numerous sieve-plate-like structures, with their appearance similar to that we had previously published for VEGF-164 in bEND5 cells [12,44], with smaller but more numerous fenestra sieve plates forming in areas void of microtubules (Figure 7A). TEM studies confirmed the induction of fenestra-rich sieve plates (Figure 7B). To determine if exposure to ouabain and inhibition of the sodium pump were linked to an activation of key signalling pathways within the membrane cytoskeleton, we first performed pharmacological blockade of several key kinases linked to the Na,K-ATPase, such as rho kinase inhibitor and PI 3-kinase inhibitors [45,46,47]. While rho kinase inhibitor Y27632 had no significant effect on induction of fenestrae, the PI 3-kinase inhibitors wortmannin and LY294002 both nearly abolished fenestra formation by either LtA or ouabain (Figure 7C,D). Further investigation demonstrated the presence of the p110α catalytic subunit and the p85 regulatory subunit of PI3-kinase in bEND5 cells. Immunoprecipitation analysis demonstrated that the association of p85 with the Na,K-ATPase was significantly elevated during fenestra induction triggered by LtA or ouabain (Figure 7F). These data suggest a role for the Na,K-ATPase in the recruitment of the class I PI 3-kinase to the sieve plate membrane cytoskeleton to trigger fenestra formation.

Finally, we decided to probe the potential relevance of these findings in vivo. Roberts and Palade demonstrated that topical application of VEGF-A to muscle could induce underlying vessel permeability and fenestration within ten minutes in the rat cremaster [17,48]. These findings suggest that VEGF-A is able to trigger a reorganisation of existing endothelial cell machinery sufficient to drive large scale fenestral pore formation. Vehicle injections had no effect on vascular permeability; however, similar to VEGF-164, we found that topical application of ouabain (a Na,K-ATPase inhibitor) resulted in a rapid (10 min) increase in vascular permeability, based on extravasation of a fluorescent 3 kDa dextran tracer (Figure 8A). TEM analysis was performed to determine the effect of the treatments on fenestra formation. The superficial microvasculature in the control cremaster muscle contained a continuous endothelium that was not fenestrated (Figure 8B). In contrast and similar to results with VEGF-164, exposure to ouabain for 10 min led to rapid attenuation of microvessels and the induction of numerous endothelial fenestrae (Figure 8D). These newly formed fenestrae were approximately 60 nm in diameter and typically had a diaphragm. Remarkably, the extent of endothelial fenestration, quantified as the % of fenestrated vessel profiles, was higher for ouabain than VEGF-164 (Figure 8E). Similar findings were obtained with another cardiac glycoside, digoxin, which is also used to inhibit the Na,K-ATPase in cardiovascular disease patients. These data identify an important regulatory role for the Na,K-ATPase in fenestra formation in vivo.

## 4. Discussion

Here we define a membrane cytoskeletal complex that regulates the formation of endothelial fenestrae. The assembly of this fodrin cytoskeleton complex is promoted by the action of the plasma membrane–actin linker protein moesin and negatively regulated by the action of the pleiotropic actin-binding protein annexin II. Both moesin and annexin II have been implicated in controlling of different aspects of actin dynamics at the plasma membrane and in endocytic processes, respectively. Although moesin’s role as a linker between the plasma membrane and actin is easier to fit into a role in linking fenestrae to the submembrane cytoskeleton, one can speculate the annexin II plays a role in the rapid change to membrane dynamics that must be altered to make the drastic changes seen upon fenestra formation.

Knockdown of fodrin expression prevented fenestra formation, demonstrating the essential nature of this cytoskeletal complex. The identification of fodrin as an essential factor for fenestra formation provides another example of how the spectrin protein family and its accessory components are utilised to create stable, functionally distinct membrane protein domains. In addition to the well-defined role of spectrin in strengthening the erythrocyte plasma membrane, fodrin, the nonerythroid spectrin, is linked to the creation of polarised plasma membrane domains in epithelium and neurons, and a Golgi spectrin regulates structural integrity and secretion in this organelle [49,50,51]. The presence of fodrin in the sieve plates could also provide a means for explaining the characteristic organisation of fenestrae into evenly spaced, linear arrays of plasma membrane pores. Each repeating tetrameric unit of the scaffold-like spectrin and its accessory proteins has been estimated to be between 35–100 nm in length in erythroid cell ghosts [52], which makes it feasible that the fodrin cytoskeletal unit could act to provide a template for pore order within the sieve plates. This idea is supported by the organisation of the cytoskeleton and fenestral pores observed in the high-pressure freeze electron tomography.

Investigators have previously searched for fenestra-associated cytoskeleton, primarily in relation to the observation that the fenestral pore is able to contract in a Ca^2+^-dependent manner, and several investigators have suggested a role for actin [24,53]. However, the observation that disassembly of actin microfilaments induces fenestra numbers has always confounded any straightforward explanation for a structural role for actin in fenestra form or function. Our data now suggest that disassembly of actin filaments using depolymerising agents such as LtA promotes the redistribution of a distinct form of actin for interactions within the fodrin-rich sieve plates. Future work will be focused on a more refined topological understanding of the fodrin membrane cytoskeleton and its relation to sieve plate and fenestral architecture.

One important molecular link between the fodrin membrane cytoskeleton and the fenestral pore is the Na,K-ATPase, which is a plasma membrane component of the fodrin complex in sieve plates. Immuno-EM data suggest that the pump may reside on or near the actual pore, making it a candidate linker between the pore diaphragm protein, PV-1, and fodrin. Studies probing direct interactions and the stoichiometry of the interacting components will be required to confirm such notions. However, there are already strong indications that the role of the sodium pump in this multiprotein complex is not simply structural. Inhibition of sodium pump activity potently triggers fenestra formation both in vitro and in vivo. Previous work has established that, in many cells, the Na,K-ATPase normally resides in caveolae; upon binding of the pump to the cardiac glycoside inhibitor ouabain, the pump is internalised and redistributes to form a distinct protein complex termed the signalosome, which in turn activates a number of downstream effectors, including Src and PI 3-kinase [38,54,55]. Accordingly, we found that triggering fenestra formation led to the formation of a new complex which included the Na,K-ATPase, fodrin, and the fenestra diaphragm protein PV-1. Presumably, the Na,K-ATPase follows a path that is similar to PV-1, which we have previously shown is rapidly internalised and shifts residence from caveolae to fenestrae following LtA induction of fenestrae [12]. The inhibition of the sodium pump and its recruitment into the sieve plate cytoskeleton were accompanied by an increased association with the regulatory unit of PI 3-kinase, and our pharmacology studies showed that PI 3-kinase activity is essential for fenestra formation. Future work is aimed at providing more resolution concerning the flow of signalling events involved in the regulation of the sieve plate protein complex assembly and its coordination with membrane trafficking and membrane fusion events required for fenestra pore formation.

The in vivo findings both validate concepts derived from the in vitro model but also raise intriguing questions about the Na,K-ATPase and cardiac glycosides. Ouabain, and more commonly digoxin, are used in the clinical setting to manage congestive heart failure; although they have pleiotropic effects, the primary mode of action is believed to be the alteration of myocyte sodium and calcium levels, leading to increased contractility [56,57]. Our findings suggest these agents may have another important mode of action, through induction of fenestra formation, which could have important clinical implications and requires further study. Another aspect of the in vivo findings that requires further investigation is based on the knowledge that ouabain is an endogenous hormone, believed to be secreted by the adrenal gland [58]. The levels of ouabain and digoxin used to stimulate in vivo fenestra biogenesis were substantially higher than normal physiological levels believed to present in the human systemic circulation, but the levels of endogenous ouabain are orders of magnitude higher in patients with conditions such as congestive heart failure, where tissue oedema is commonly observed [59]. Elevation of ouabain also leads to changes to the vascular barrier in other experimental settings, such as in the cerebral vasculature, where the blood–brain barrier is rapidly compromised [60]. Based on our findings, a role for fenestra induction in the physiological, pathological, and pharmacological settings associated with ouabain or other cardiac glycosides deserves further consideration.

Finally, changes in fenestra number, size, and the presence or absence of a diaphragm, have been predicted to have significant impact on the function of many organ systems [61], though little experimental evidence has been provided until recently. Stan et al. have demonstrated that removal of the diaphragm from fenestrae through targeted endothelial cell deletion of the PV-1 gene leads to massive leakage of plasma proteins, oedema, and subsequent enteropathy [62]. These findings highlight the consequences of altering the porosity and selectivity of the endothelial barrier. We have now described several methods by which the barrier function of the endothelium can be altered, including alteration of fenestra pore size [12], reduction in fenestra number (inhibition of moesin or fodrin), or induction of fenestra number (inhibition of annexin II or the sodium pump). Our findings provide a foundation for a conceptual framework that links alterations in fenestra pore function directly to changes in the assembly of a fodrin membrane cytoskeletal complex, while also providing molecular tools for use in bettering our understanding of the liabilities and opportunities associated with manipulating the fenestra.

## Figures and Tables

**Figure 1 cells-09-01387-f001:**
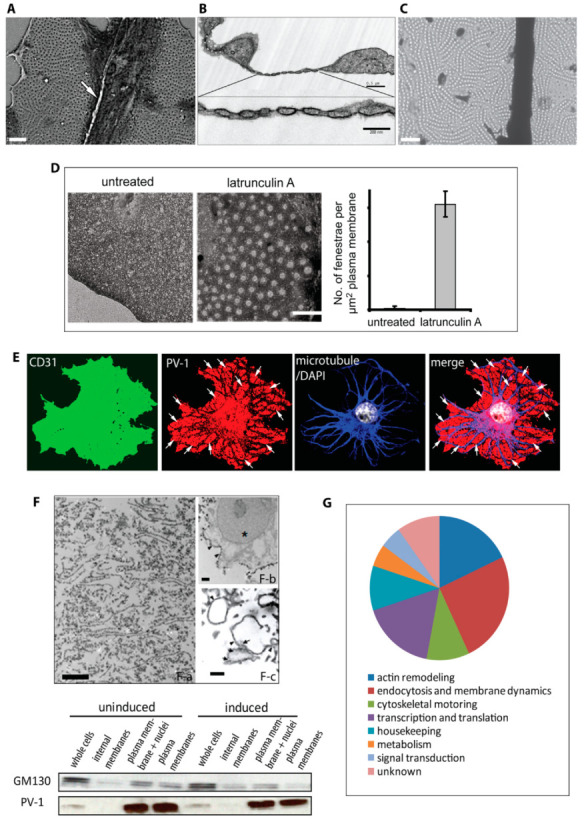
Fenestra morphological and proteomic characterisation. The pores are present in the highly attenuated cell periphery in clusters devoid of most organelles and cytoskeleton, termed sieve plates (**A**) SEM image (arrow, microtubule; scale bar, 1 μm). Fenestrae form in areas of the cell where the apical–basal plasma membrane distance is as little as 40 nm. The pore diameter is approximately 60 nm, and it is spanned by a pinwheel-like diaphragm. Substances that traverse the pore never encounter the contents of the cytoplasm. (**B**) Thin-section TEM image (arrows, fenestral diaphragm; top scale bar, 1 μm; bottom scale bar, 200 nm). In the sieve plates, fenestrae are highly organised in linear arrays. (**C**) Wholemount TEM image (scale bar, 1 μm). (**D**) Fenestra formation induced with LtA is assessed by wholemount TEM.. Quantification of 36 TEM images revealed a >100-fold of fenestrae induction over the untreated condition. Results are representatives from triplicate experiments. Error bars, SEM; scale bar, 0.5 μm. (**E**) Fenestra formation is detected by immunolabelling for the diaphragm protein PV-1, while the continuity of the bEND5 plasma membrane is indicated by immunostaining for CD31. The sieve plates are readily distinguishable with PV-1 staining at the cell periphery and are demarcated by microtubules, as shown in the merged image (arrows, sieve plate). (**F**) A silica-based affinity isolation method was used to enrich plasma membrane. (**F-a**) Scale bar, 2 μm. The electron micrographs depict isolated fenestrated membrane, whilst the immunoblot confirms the enrichment of PV-1, a marker for the fenestrated plasma membrane, versus the Golgi marker GM130. The nucleus remnants are depicted in (**F-b)** and are denoted by a star (scale bar, 0.5 μm), whilst maintaining the structural integrity of membranous microdomains such as fenestrae. (**F-c**) Fenestrae in LtA-induced bEND5 cells: fenestrae en face are denoted by arrows, and fenestrae in cross-section are denoted by arrowheads. Scale bar, 0.5 μm. (**G**) Two-dimensional gel electrophoresis and mass spectrometry were used to compare the plasma membrane protein profiles of induced and noninduced bEND5 cells; candidate proteins were obtained by a subtractive proteomic analysis. More than half of the proteins identified were involved in cytoskeletal or membrane dynamics.

**Figure 2 cells-09-01387-f002:**
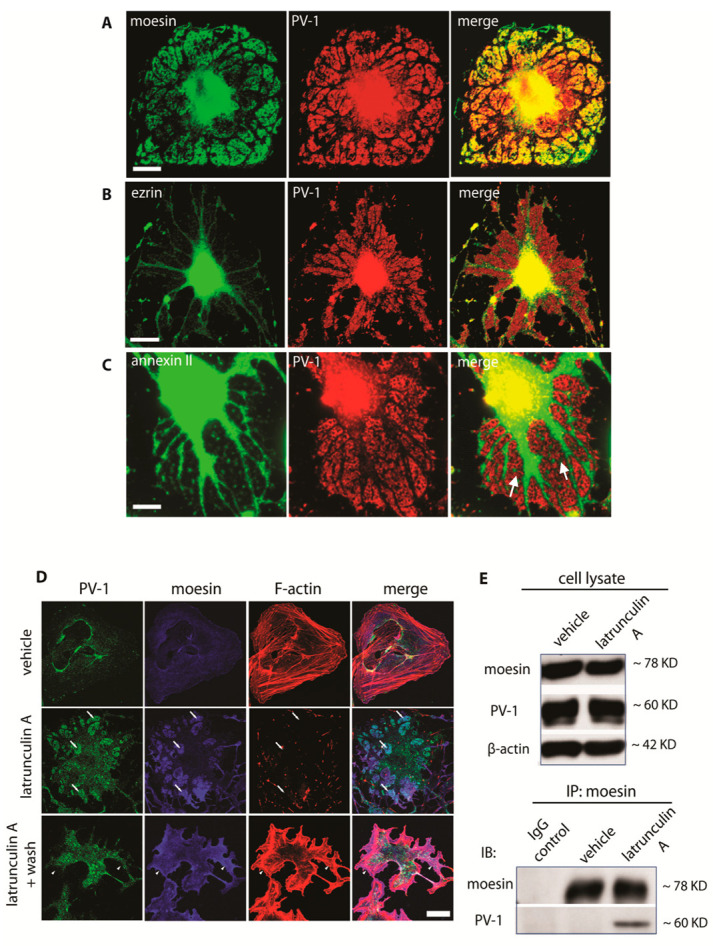
Moesin and annexin II are in fenestral sieve plates (see also Appendix A). (**A**) Moesin was detected in the LtA-induced sieve plates, and it colocalised with PV-1. Arrows, fenestral sieve plates; scale bar, 10 μm. (**B**) The ERM family member ezrin was excluded from the sieve plates. Scale bar, 10 μm. (**C**) Annexin II did not colocalise with PV-1 but was localised to small islands within the sieve plates (arrows); scale bar, 10 μm. (**D**) Upon disassembly of actin microfilaments following exposure to LtA, both moesin and PV-1 were redistributed into the distinct sieve plates (arrows). Following LtA washout, fenestrae rapidly disappeared and PV-1, moesin, and F-actin were redistributed to distinct areas of the cell periphery (arrow heads). Scale bar, 10 μm. (**E**) Immunoprecipitation with a moesin antibody demonstrated that moesin and PV-1 are only in a protein complex following fenestra induction.

**Figure 3 cells-09-01387-f003:**
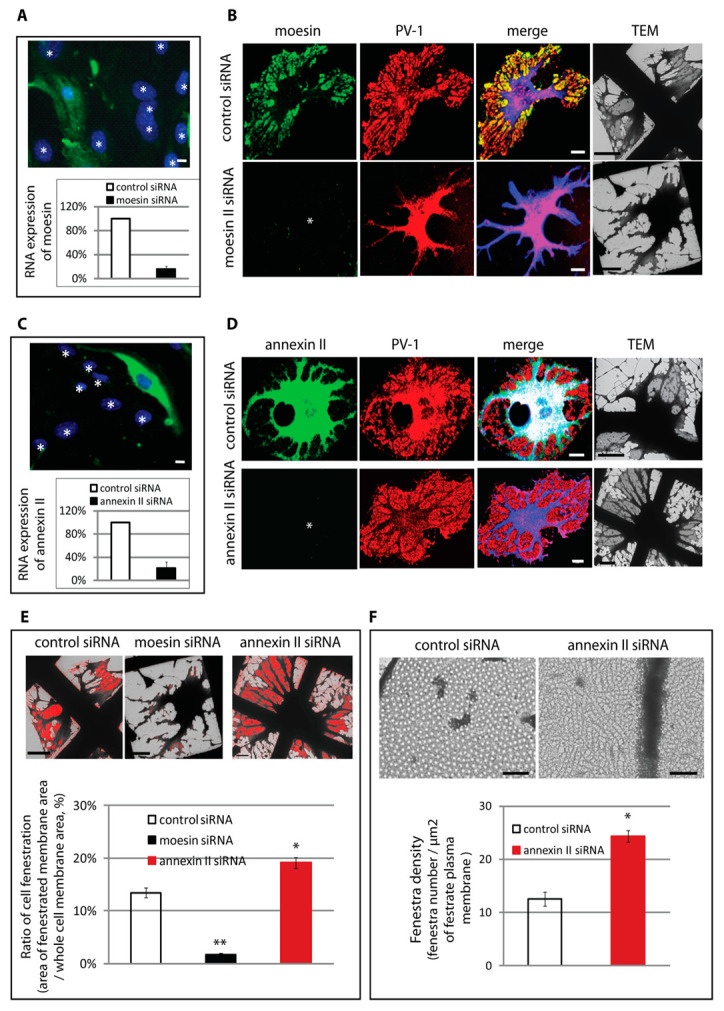
Moesin and annexin II differentially regulate fenestra formation. (**A**) Down-regulation of moesin by siRNA showed by immunostaining and qPCR analysis. The result was the average of 3 experiments (*n* = 3); scale bar, 20 μm. (**B**) bEND5 cells transfected with control siRNA or moesin siRNA were induced with LtA for 3 h. The cells were fixed and immunostained for moesin, PV-1, and β-tubulin, before being prepared for correlative light–electron microscopy. In control siRNA transfected cells, moesin and PV-1 colocalised in sieve plates, while moesin knockdown resulted in the failure of PV-1 redistribution to sieve plates and the cells maintained their normal stellate morphology. Wholemount TEM revealed the lack of fenestrated plasma membrane in moesin knockdown cells. (scale bars, 10 μm in immunofluorescence images and 5 μm in TEM images). (**C**) Down-regulation of annexin II by siRNA showed by immunostaining and qPCR analysis. The result was the average of 3 experiments (*n* = 3); scale bar, 20 μm. (**D**) bEND5 cells transfected with control siRNA or annexin II siRNA were induced with LtA for 3 h. The cells were fixed and immunostained for annexin II, PV-1, and β-tubulin, before being prepared for correlative light–electron microscopy. Annexin II depletion resulted in an increased formation of fenestral sieve plates which were revealed by both immunofluorescence staining and wholemount TEM analysis (scale bars, 10 μm in immunofluorescence images and 5 μm in TEM images). (**E**) Quantification of the area of fenestrated plasma membrane showed that moesin knockdown reduced the formation of fenestral sieve plates (>90%), while annexin II depletion significantly increased the area of fenestrated plasma membrane by 25%. Error bars, SEM; * *P* < 0.01; ** *P* < 0.001; *n* ≥ 30. (**F**) Annexin II depletion resulted in increased density of fenestra per μm^2^ of plasma membrane. Scale bar, 0.5 μm. Error bars, SEM; * *P* < 0.001; *n* ≥ 20. Number of fenestrae and the area of fenestrated plasma membrane were measured using Pro-Plus 6.1 image software.

**Figure 4 cells-09-01387-f004:**
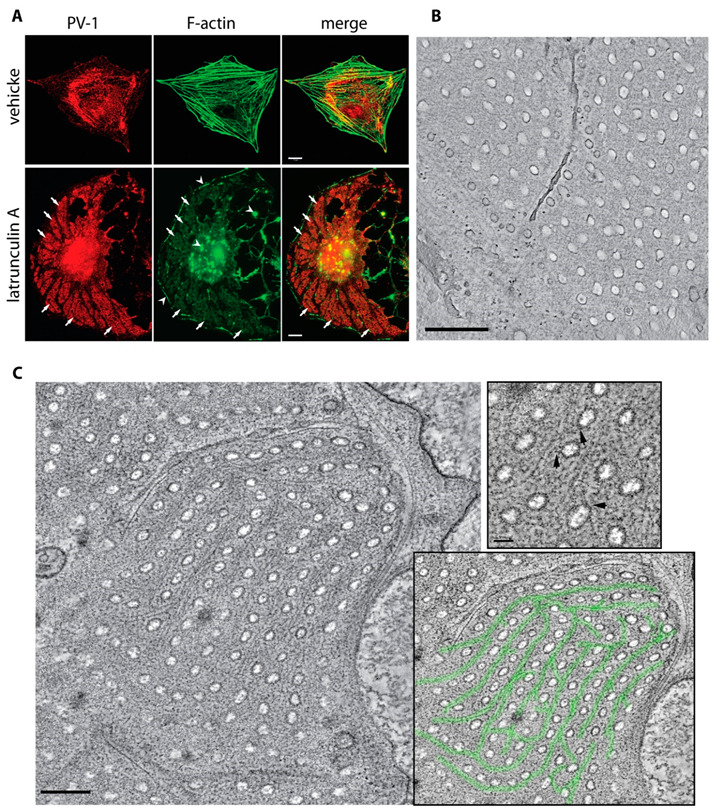
Organisation of the actin cytoskeleton in the fenestral sieve plates. (**A**) bEND5 cells treated with vehicle or induced with LtA for 3 h were fixed and immunostained for PV-1, and F-actin was revealed using fluorescence phalloidin. Although F-actin filaments were disassembled following induction, weaker phalloidin staining was consistently present in PV-1-positive fenestral sieve plates (arrows), suggesting the presence of short actin filaments that were beyond the resolution of light microscopy. Scale bar, 10 μm. (**B**) No discernible cytoskeletal structure was observed in tomographic projections from chemically fixed bEND5 cells (scale bar, 1 μm) (see also Video S1A), but a lattice-like cytoskeleton (**C**) was consistently observed in tomographic projections from high-pressure frozen and freeze substituted bEND5 cells (left panel, highlighted in green; scale bar, 500 nm) (see also Video S1B). A cytoskeleton that runs proximal and parallel to the linear arrays of fenestral pores in sieve plates is shown at high magnification. Links between the actin cytoskeleton and the rim of the fenestral pore were sometimes distinguishable (arrows in top right panel of (**C**); scale bar 50 nm). The network in an image of the sieve plate is traced in the bottom-right panel of (**C**).

**Figure 5 cells-09-01387-f005:**
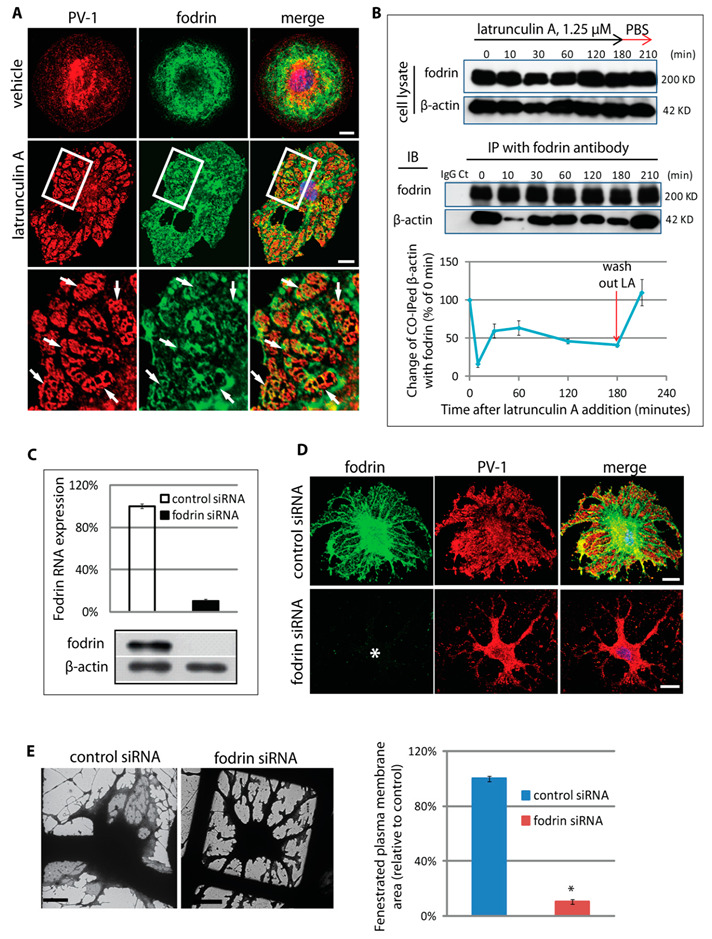
Fodrin is part of the sieve plate cytoskeleton and is required for fenestra formation. (**A**) Following fenestra induction, fodrin was reorganised into the PV-1-positive sieve plates (arrows, fenestral sieve plates; scale bar, 10 μm). (**B**) Immunoprecipitation of fodrin revealed a dynamic interaction with β-actin during the time-course of fenestra induction, consistent with a redistribution of β-actin from large microfilament networks into the sieve plate fodrin cytoskeleton. The result of every time point is the average of 3 independent experiments (*n* = 3). (**C**) Fodrin depletion in cells was achieved by siRNA transfection. Quantification based on triplicate experiments of fodrin in siRNA showed 85% reduction of mRNA, with a similar level of reduction in fodrin protein shown by Western blot. The result of is the average of 3 independent experiments (*n* = 3). (**D**) Immunofluorescence staining for PV-1 showed similar appearance to moesin knockdown, with PV-1 accumulation in the organelle-rich region associated with the microtubule arbours (asterisk, fodrin depleted bEND5 cell; scale bar, 10 μm). (**E**) Light–electron microscopy correlated bEND5 cells confirmed >90% reduction in fenestrated plasma membrane in fodrin siRNA-treated cells. Left panel, cell phenotype revealed by TEM; scale bar, 2 μm. Right panel, quantification of fenestrated membrane area from TEM images. Error bars represent ± SEM. * *P* < 0.001; *n* ≥ 30.

**Figure 6 cells-09-01387-f006:**
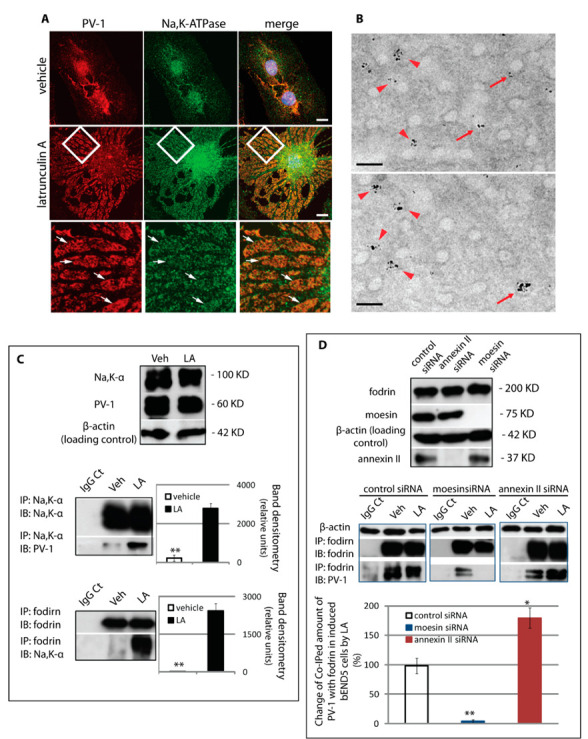
Na,K-ATPase is a transmembrane component of the fenestral sieve plates. (**A**) Following fenestra induction, Na,K-ATPase was enriched in the PV-1-positive sieve plates (arrows, fenestral sieve plates; scale bar, 10 μm). (**B**) Pre-embedding immunolabelling for Na,K-α subunit using a gold-conjugated Fab fragment and silver enhancement in thin sections cut along the plane of the cell monolayer revealed Na,K-ATPase labelling in the sieve plate area, localised either adjacent to the pore rim (arrowheads) or often on the rim of fenestral pores (arrows). Scale bar, 100 nm. (**C**) Immunoprecipitation revealed an increased association of Na,K-α and PV-1 proteins following induction of fenestra formation. The Na,K-α subunit also co-immunoprecipitated with fodrin in fenestrated cells. Mean and standard deviation of three experiments are shown. ** *P* < 0.0001. (**D**) The interaction between fodrin and PV-1 was regulated by moesin and annexin II. Expression of moesin or annexin II was successfully knocked down by siRNA. The bEND5 cells transfected with control siRNA, moesin siRNA, or annexin II siRNA were treated with vehicle or induced to form fenestrae. Immunoprecipitation with a fodrin antibody demonstrated that, in bEND5 cells that were fenestrated, the amount of co-immunoprecipitated (co-IPed) PV-1 with fodrin diminished upon moesin depletion but increased upon annexin II depletion. Mean and standard deviation of three experiments are shown (*n* = 3). * *P* < 0.01; ** *P* < 0.0001. Veh, vehicle; LA, latrunculin A.

**Figure 7 cells-09-01387-f007:**
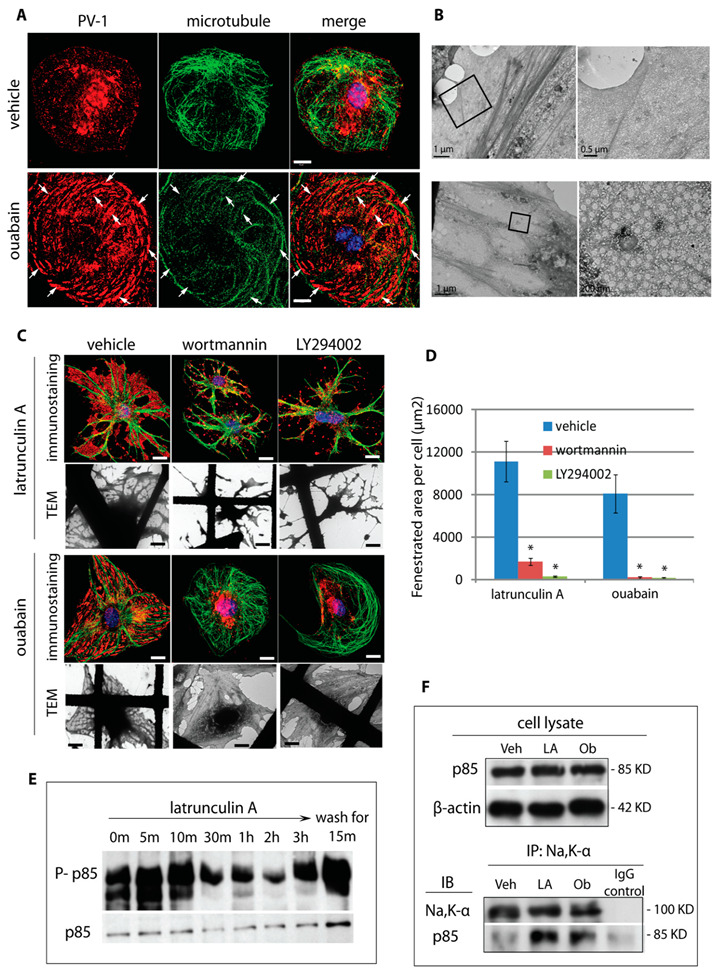
Inhibition of Na,K-ATPase pump function by ouabain induces fenestra formation. (**A**) bEND5 cells were treated with vehicle (PBS) or ouabain at 200 μM for 24 h. The cells were fixed and immunostained for PV-1 and β-tubulin. Numerous, relatively small sieve plates (arrows) appeared between microtubules (merge panels) in ouabain-treated cells, just as observed in LtA-treated cells. Scale bar, 10 μm. (**B**) Wholemount TEM analysis confirmed ouabain-induced fenestra formation; right panels show higher magnification view; arrows, fenestral sieve plates. (**C**) Blockade of PI 3-kinase signalling inhibited fenestra formation. Typical microtubule-delineated sieve plates were induced by LtA or ouabain, as shown by immunostaining for PV-1 (red) and β-tubulin (green); DAPI, blue. LtA and ouabain induction of fenestra formation was inhibited by co-incubation with the PI 3-kinase inhibitors, wortmannin (1 μM), or LY294002 (10 μM). The reduction of fenestral sieve plates was confirmed by wholemount TEM (lower panels). Arrows, fenestral sieve plates; scale bar, 10 μm. (**D**) Quantification of fenestrated membrane area from wholemount TEM images showed a near complete inhibition of fenestra formation by wortmannin or LY294002. Error bars represent ± SEM; *n* ≥ 15; * *P* < 0.0001. (**E**) PI 3-kinase subunits p85 and p110α were present in bEND5 cells, and levels were unchanged following fenestra induction. However, phospho-p85 decreases during fenestra induction and is lowest as fenestrae formation peaks. (**F**) Immunoprecipitation with an antibody to Na,K-α demonstrated an increased association of p85 and the sodium pump following induction of fenestrae by LtA or ouabain.

**Figure 8 cells-09-01387-f008:**
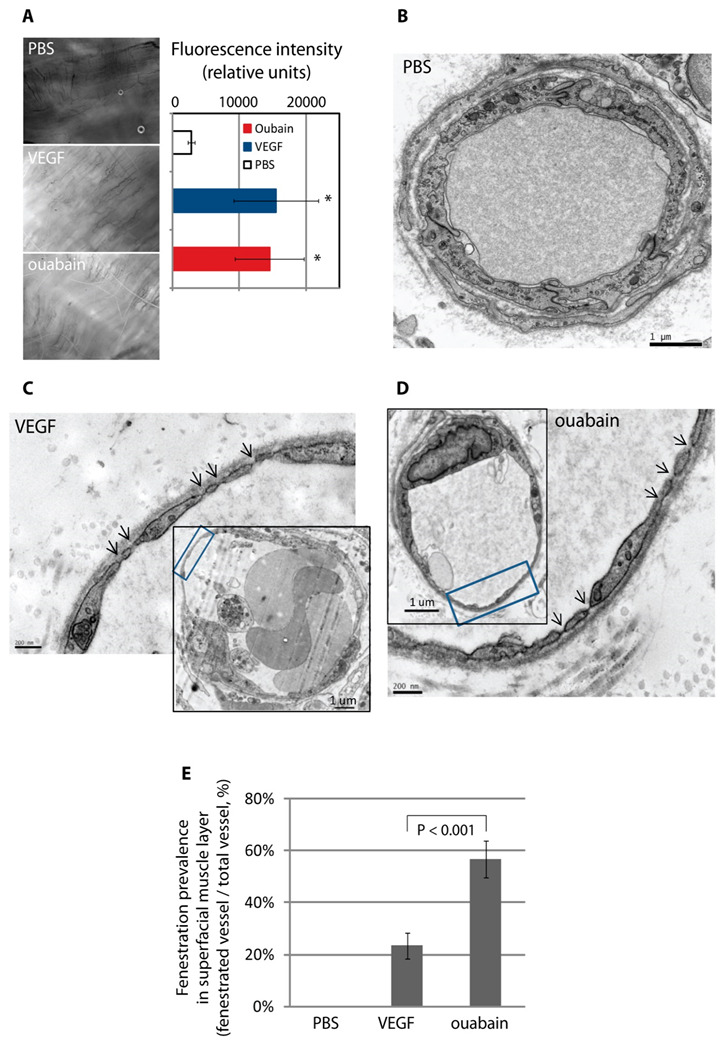
Inhibition of the Na,K-ATPase pump induces fenestra formation in vivo. (**A**) Ouabain increased vascular permeability in rat cremaster muscle. PBS (100 μL), VEGF-164 (75 ng in 100 μL), and ouabain (200 μM, 100 μL) were topically applied on surgically exposed rat cremaster muscle. Leakage of a fluorescent systemic tracer (fixable 3kD fluorescent dextran) was significantly increased 10 min after exposure to VEGF-164- or ouabain-treated samples. Error bars represent ± SEM; *n* = 3; * *P* < 0.01. (**B**–**D**) ouabain induced fenestra formation in vivo. Rat cremaster tissue subjected to topical application of PBS (100 μL), VEGF-164 (75 ng in 100 μL) or ouabain (200 μM, 100 μL) was further processed for TEM analysis. Microvessels in PBS-treated samples were not fenestrated (**B**), whilst microvessels with extremely attenuated vessel walls and abundant fenestrae were found in VEGF-164-or ouabain-treated samples ((**C**) and (**D**); arrows, fenestrae). (**E**) The number of fenestrated vessel profiles was determined and expressed as a percentage of the total number of vasculatures. The incidence of fenestration was significantly higher in ouabain-treated cremaster tissue compared to control and even to VEGF-164-treated samples. Mean and standard deviation of three independent experiments are shown; *P* < 0.001.

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
