# Peer review of "A Na,K-ATPase–Fodrin–Actin Membrane Cytoskeleton Complex is Required for Endothelial Fenestra Biogenesis"

_cells, 2020, doi:10.3390/cells9061387_

Round 1
Reviewer 1 Report
The manuscript by Meihua Ju et al. entitled “A Na,K-ATPase-Fodrin-Actin Membrane Cytoskeleton Complex is Required for Endothelial Fenestra Biogenesis” describes a novel molecular regulation of fenestra biogenesis by moesin and Annexin II and discover a molecular complex involving Na, K-ATPase-Fodrin-Actin that promotes fenestra formation. The findings are very interesting and of great physiological relevance. To this reviewer the nontraditional roles for the actin cytoskeleton and Na, K, ATPase is intriguing. This is a very well executed study.
Here are some minor comments:
- Some areas in the results section says “data not shown”. Please leave out these points.
For example, there is no need to explain the alternative cell model, for figure 2B just discuss Ezrin for which data is shown and leave out radixin and merlin.
- Figure A1A is not included in the files this reviewer received. The supplemental document only has one movie.
- Figure 1 immunoblots are not discussed properly in the text. It is not clear if it belongs to figure 1F or 1G.
- Figure 4A label – spell check.
- Sentences 428 – 430, make sure it says p110 and p85 subunits of PI3K
- Please include sample size wherever possible in the figure legend.
- None of Immunoblots and Immunoprecipitations have ladder lane. This reviewer is fine with cropped blots as long as I get to see uncropped version of at least one of the presented blots.
- The authors do not discuss how moesin is linked to fodrin-actin cytoskeletal remodeling during fenestra biogenesis. Perhaps they can make some speculation in the discussion? Similarly negative regulation by Annexin II must also be discussed.
Author Response
- Thank you for the comments. Data not shown and related references to merlin and radian have been removed
- We have clarified the Figure 1 legend to denote the intent of the distinct data sets
- There was a glitch in uploading the videos. We are working with the editors to rectify
- We have clarified PI3 kinase as the p110/85 referral
- We have not placed an uncrossed blot in the manuscript, partly because the data are inaccessible (Covid) but can assure the reviewer that the blots were repeated a minimum of three times
- Though early days, we have tried to speculate further on the role of moesin, fodrin and annexin II in fenestra formation, referring to their known roles with structural support and membrane dynamics
Again, thank you for making the manuscript clearer
Reviewer 2 Report
In the present study, Ju et al investigated the Fenestra formation and analyzed underlying mechanisms in endothelial cells. They report that fodrin-actin submembrane cytoskeleton, regulated by moesin, is essential for fenestra biogenesis. Moreover, they provide in vitro and in vivo evidence that Na/K ATPase signaling is required for this response.
This is a well-designed and well-conducted study. Experiments are carefully done. Results are interesting. They are appropriately presented and well discussed. This study contributes to the understanding of the complex composition and regulation of those transcellular plasma membrane pores in vascular endothelia.
Some issues however, indicated below, could be addressed, in order to improve the overall quality of this work.
Comments
Figure 1. : The information provided by panels F and G should be documented in a clearer manner.
Figure 2. : Panel B/C: The conclusion stated for erzin staining data should be analyzed in a more comprehensive way. In addition, description of annexin II data need further confirmation.
Figure 4. : Since only actin microfilament staining has been analyzed, the term cytoskeleton should be replaced with actin cytoskeleton. In addition, in panel C some details need further clarifications. Indeed, the links between actin cytoskeleton and the rim of the fenestral pore are difficult to be distinguished. Moreover, it is unclear which is the additional information provided by the bottom right panel. Please, either specify or remove.
Figure 7. : Authors indicate that Rho signaling is not involved in fenestra formation linked to Na/K ATPase activation, while they demonstrate the involvement of PI3K by using two distinct PI3K-inhibitors on fenestra formation. Since –besides Rho signaling- PI3K downstream signaling may as well regulate early actin cytoskeleton rearrangements, which in turn seem to be linked (this work) to fenestra formation, it is not obvious why microtubules but not actin was included in this analysis. Authors should consider addressing this important issue.
Author Response
Thank you for the helpful comments.
- We have clarified Figure 1 as requested, adding to the legend additional information
- We have elaborated on ezrin and its lack of clear involvement. What is interesting is that besides its very similar role to moesin's, it appears to be functioning on organelles or membranes with the microtubule bundles.
- We have clarified about the actin cytoskeleton, rather than just cytoskeleton.
- We have not explored microtubules directly. We know that they are excluded from fenestral sieve plates, so did not prioritise exploration of their role.
Thanks again for helping to improve the manuscript
David